

# Differences in electromyographic activity of biceps brachii and brachioradialis while performing three variants of curl

Giuseppe Marcolin[1], Fausto Antonio Panizzolo[2], Nicola Petrone[3], Tatiana Moro[1], Davide Grigoletto[1], Davide Piccolo[4] and Antonio Paoli[1]

[1] Department of Biomedical Sciences, University of Padova, Padova, Italy
[2] John A. Paulson School of Engineering and Applied Sciences, Harvard University, Cambridge, United States of America
[3] Department of Industrial Engineering, University of Padova, Padova, Italy
[4] School of Human Movement Sciences, University of Padova, Padova, Italy

## ABSTRACT

**Background**. Dumbbell curl (DC) and barbell curl in its two variants, straight (BC) or undulated bar (EZ) are typical exercises to train the elbow flexors. The aim of the study was to verify if the execution of these three variants could induce a selective electromyographic (EMG) activity of the biceps brachii (BB) and brachioradialis (BR).

**Methods**. Twelve participants performed one set of ten repetitions at 65% of their 1-RM for each variant of curl. Pre-gelled electrodes were applied with an inter-electrode distance of 24 mm on BB and BR. An electrical goniometer was synchronously recorded with EMG signals to determine the concentric and eccentric phases of each variant of curl.

**Results**. We detected higher activation profile of both BB ($P < 0.05$) and BR ($P < 0.01$) during the EZ compared to the DC. Higher levels of activation was found during the concentric phase for only the BR performed with an EZ compared to DC ($P < 0.001$) and performing BC compared to DC ($P < 0.05$). The eccentric phase showed a higher activation of the BB muscle in EZ compared to DC ($P < 0.01$) and in BC compared to DC ($P < 0.05$). The BR muscle showed a higher activation performing EZ compared to DC ($P < 0.01$).

**Discussion**. The EZ variant may be preferred over the DC variant as it enhances BB and BR EMG activity during the whole range of motion and only in the eccentric phase. The small difference between BC and EZ variants of the BB and BR EMG activity makes the choice between these two exercises a matter of subjective comfort.

# INTRODUCTION

Resistance training has fundamental importance in improving athletic performance as it allows the increase of muscular strength, power and speed (*Kraemer & Ratamess, 2004*). In strength athletes such as bodybuilders, resistance training plays a key role as it induces muscle hypertrophy. Therefore, to maximize muscle hypertrophy, bodybuilders construct

Corresponding author
Giuseppe Marcolin,
giuseppe.marcolin@unipd.it

training programs which involve exercises isolating specific muscles with different exercise variants or different ranges of movement, with the aim to increase muscle activity.

These training approaches have been investigated in both lower and upper body exercises which target specific muscle groups. For instance, the effect of feet position on the electromyographic (EMG) activity of quadriceps muscles has been previously reported (*Boyden, Kingman & Dyson, 2000*; *Signorile, Kwiatkowski & Caruso, 1995*), as well as the effect of stance width on the activity of the gluteus maximus during a back squat (*Paoli, Marcolin & Petrone, 2009*). In addition, EMG of the shoulder and trunk muscles during different variations of the lat pull down exercise was investigated (*Lusk, Hale & Russell, 2005*; *Signorile, Zink & Szwed, 2002*), showing that maximal activation of the latissimus dorsi was obtained when performing the exercise with an anterior wide grip. The effect of grip width and forearm pronation/supination on upper body muscles while performing the flat bench-press exercise was also explored and it was found that small changes in muscle activity were associated with changes in grip width (*Lehman, 2005*).

Previous studies have found that changes in the technical execution of an exercise could selectively influence muscle activity though there is very little information available focusing on exercises targeting the elbow flexors (biceps, brachialis and brachioradialis). Oliveira and Goncalves (*Oliveira & Gonçalves, 2009*) found that different body posture influenced the demand of neuromuscular, cardiovascular and sensorial responses. Another study (*Oliveira et al., 2009*) reported that standing dumbbell curls and sitting dumbbell curls with the trunk inclined backwards were recommended for biceps force improvement. This was due to an overall higher neuromuscular effort in the whole range of motion while in the dumbbell preacher curls the activation of the biceps was maximal only for elbow angles close to full extension.

As reported in bodybuilder manuals (*Hatfield, 1993*) it is well accepted that two of the most employed dumbbell curls, the incline curls and the hummer curls, pre-stretch the biceps long head and enhance the involvement of the brachialis, respectively. On the other end, a very popular barbell curl named the Scott curls, unload the long head of the biceps placing greater overloading on the short head (*Hatfield, 1993*). Furthermore, performing barbell curls with a reverse handgrip resulted in an increase in brachioradialis activation (*Hatfield, 1993*). These exercises aim at inducing muscular hypertrophy of the elbow flexors, and can be performed using a straight bar, an undulated bar (named ''EZ'') or dumbbells.

To extend the findings of previous work and considering that there is no clear consensus in the employment of the straight bar rather than the undulated bar or the dumbbells when the aim is to increase the EMG activity of the biceps brachii and brachioradialis, the purpose of the present study was to investigate if the execution of barbell curls with straight bar (BC), barbell curls with undulated bar (EZ) and alternate dumbbell curls (DC) affect the EMG activity of elbow flexors. Furthermore, we hypothesized that the EZ variant will induce a higher EMG activity of the biceps brachii and brachioradialis due to the almost semiprone forearm position which would increase the activation of the two muscles (*Basmajian & De Luca, 1978*).

## MATERIALS AND METHODS

Twelve male participants (age $25 \pm 5$ years, body mass $77 \pm 9$ kg, height $183 \pm 6$ cm) with at least 3 years resistance training experience (three resistance training sessions per week) were recruited in the present study. All participants had to habitually train with both dumbbell and barbell curls and were right hand dominant. At the time of the study the participants did not present any pathology of the shoulders, elbows and wrists and they were free from neuromuscular diseases. A detailed description of the experimental procedures was given to each participant and informed consent was obtained. The experimental protocol adhered to the principles of the 1975 Helsinki Declaration and was approved by the ethical committee of the Department of Biomedical Sciences, University of Padova (HEC-DSB12/16).

Muscle activity was recorded by means of a PDA Pocket EMG (BTS Bioengineering, Milan, Italy). Device resolution was 16 bit, weight 300 g and dimensions $145 \times 95 \times 20$ mm. The sampling frequency was set to 1 kHz to avoid aliasing phenomena. Muscles analyzed were the biceps brachii (BB) and the brachioradialis (BR) of the right limb of each participant. Ag/AgCl pre-gelled electrodes were applied with an inter-electrode distance of 24 mm. Skin preparation, sensor location and orientation on the muscles were in accordance with Hermens and colleagues (*Hermens et al., 2000*). In order to determine the concentric and the eccentric phases of the exercise, an electrical goniometer (Biometrics LTD, Newport, UK) was placed on the right elbow of each participant and recorded at 1 kHz (synchronously with EMG signals).

A standing posture was maintained during the performance of the three exercises: BC, EZ, DC. The one repetition maximum (1-RM) was determined for each of the three exercises by means of a submaximal estimation method during three separate days as detailed in Brzycki (*Brzycki, 1993*). The experimental data collection were carried out in the biomechanical laboratory and performed on the same day for all participants. Each participant performed a standardized warm-up consisting of 12 repetitions of alternate dumbbell curls at 40% of his 1-RM. The participants were then asked to execute one set of 10 repetitions for each of the three variants in a randomized order. A metronome was employed to impose the same time of execution among sets and among participants. The metronome was set so that each repetition duration was 3 s. The load employed corresponded to 65% of the 1-RM. The rest between sets was fixed at 4 min to allow an adequate recovery. During the execution of the repetitions participants had to assume a standing posture and to follow the rhythm imposed by the metronome as much as possible. Trunk movements together with knee flexions were visually monitored by the researchers to avoid possible cheating. For better clarity a scheme of the experimental protocol is reported in Fig. 1.

To obtain a linear envelop, EMG interference signals were first rectified around their mean value, then integrated with a moving window of 200 ms, and finally smoothed with a 4th order Butterworth low pass filter set at 5 Hz. The concentric phase was defined from elbow maximal extension to maximal flexion, the eccentric phase was defined from elbow maximal flexion up to maximal extension, obtained by means of the electrical goniometer. The first and last repetitions (out of 10) were not selected for analysis due

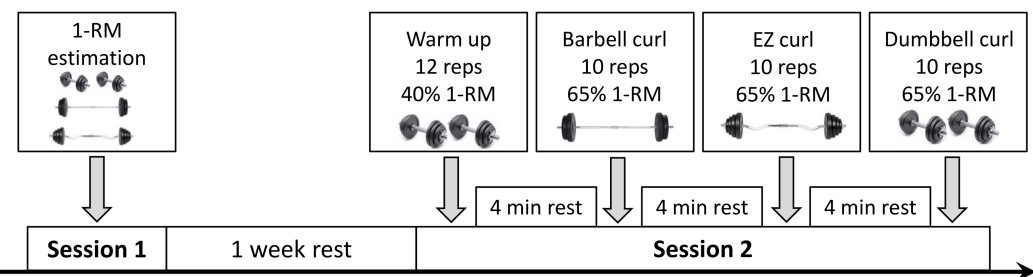

**Figure 1** **Experimental protocol.** Graphical representation of the experimental protocol.

to the inconsistencies in technique (*Paoli, Marcolin & Petrone, 2009*; *Paoli, Marcolin & Petrone, 2010*). For each participant and each experimental condition, the mean EMG activity was computed for each repetition (relative to the concentric, eccentric phase and whole movement) and an average of the eight repetitions was then calculated and reported for each of the three exercises, both for BB and BR.

Friedman non-parametric test for repeated measurements was used to compare the three exercises. Significant level was set at $P < 0.05$. If a statistically significant difference was found, Dunn's multiple comparison test was employed. Data analysis was performed by means of the software package GraphPad Prism version 4.00 for Windows (GraphPad Software, San Diego, CA, USA). Statistical effect size was calculated with the G*Power 3.1.5 software (*Faul et al., 2007*).

## RESULTS

The time of execution considering all the repetitions of the three variants was $3.16 \pm 0.67$ s. In particular, the concentric movement duration was $1.44 \pm 0.26$ s while the eccentric movement was $1.72 \pm 0.47$ s. The range of motion (ROM) mean of the right elbow (Fig. 2) recorded in the three variants of curl (BC = $117.3° \pm 10.9°$; EZ = $119.9° \pm 13.7°$; DC = $123.1° \pm 12.3°$) was found to be statistically different ($P = 0.0087$). However, a post hoc test showed only a greater ROM in BC with respect to DC ($P < 0.05$, ES = 0.46). In Fig. 3 we reported an example of representative EMG data (bicep activity of 1 subject in the three variants of curl). Considering the whole range of motion (Fig. 4A) significant differences were observed in the EMG activity of the BB ($P = 0.0204$) and BR ($P = 0.0023$). A post hoc test showed a higher activation during the EZ variant with respect to the DC both for BB ($P < 0.05$, $ES = 0.30$) and BR ($P < 0.01$, $ES = 0.51$). No differences were detected between EZ and BC for both muscles investigated.

The concentric movement (Fig. 4B) showed statistically significant differences for the BR ($P = 0.0009$) but not for the BB. A post hoc test showed a higher activation of the BR during EZ variant with respect to DC ($P < 0.001$, $ES = 0.48$) and of the BC with respect to DC ($P < 0.05$, $ES = 0.36$).

The eccentric movement (Fig. 4C) showed a difference in the EMG activity for the BB ($P = 0.0014$). Specifically, muscle activity was higher in the EZ variant compared to the DC

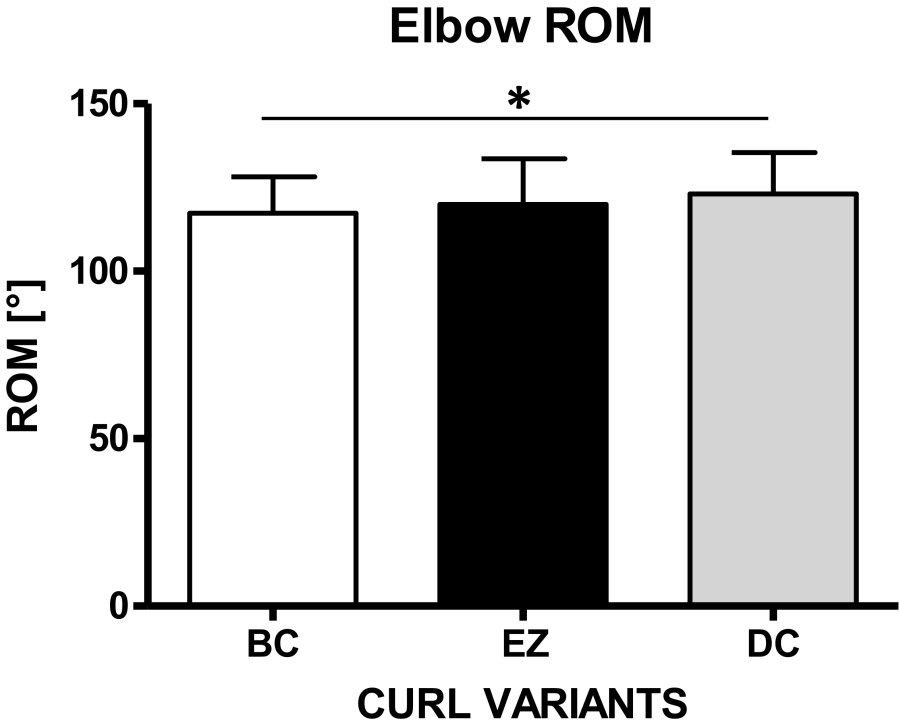

**Figure 2  Range of motion.** Mean values of the range of motion (ROM) recorded in three variants of the curl (*$p < 0.05$).

variant ($P < 0.01$, $ES = 0.45$) as well as in the BC variant compared to the DC ($P < 0.05$, $ES = 0.28$).

BR muscle showed statistically significant differences during the eccentric phase ($P = 0.0038$) with a higher activation during the EZ variant in comparison to the DC variant ($P < 0.01$, $ES = 0.66$).

## DISCUSSION

The enhancement of the activity of specific muscles with different exercise variants and different ranges of motion have been extensively examined for both the lower limb muscles (*Boyden, Kingman & Dyson, 2000*; *Paoli, Marcolin & Petrone, 2009*; *Signorile, Kwiatkowski & Caruso, 1995*) and trunk muscles (*Lehman, 2005*; *Lusk, Hale & Russell, 2005*; *Marcolin et al., 2015*; *Paoli, Marcolin & Petrone, 2010*; *Signorile, Zink & Szwed, 2002*). Nevertheless, few studies have investigated the EMG activity of the arm and forearm muscles while performing resistance exercises (*Oliveira & Gonçalves, 2009*; *Oliveira et al., 2009*). To the best of our knowledge, this was the first study to assess the influence of different variants of the curl exercise on the level of activation of the BB and BR muscle.

In the present work, the EZ variant exhibited the highest level of EMG activity for both BB and BR. However, significant differences were observed only in comparison with the DC variant. The concentric phase analysis of the BB showed that the EZ variant induced the highest level of EMG activity (+7% with respect to BC and +11% with respect to DC)

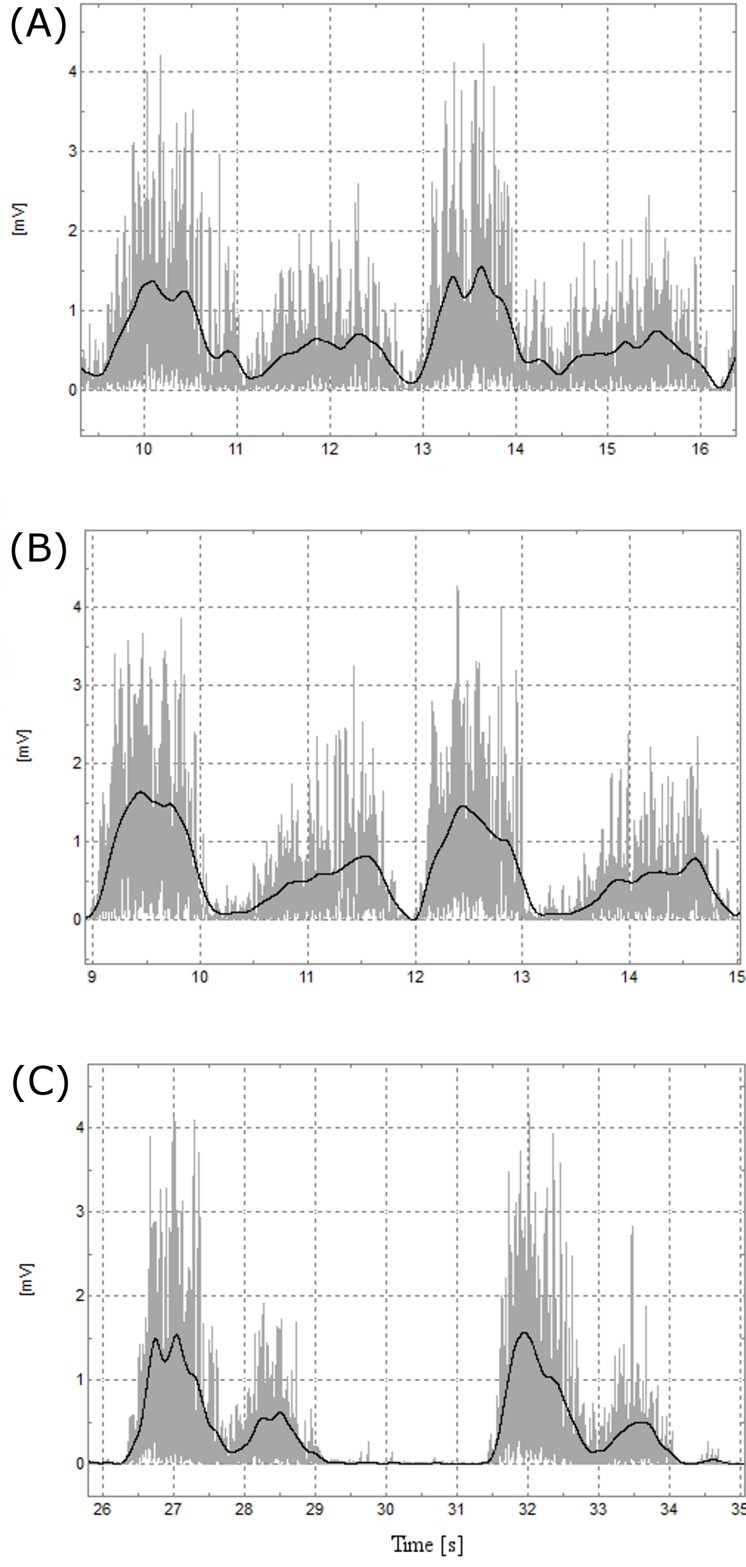

**Figure 3  Representative EMG data.** Representative biceps brachii EMG data of two repetitions of the three variants of curl. From top to bottom: (A) BB, (B) EZ and (C) DC.

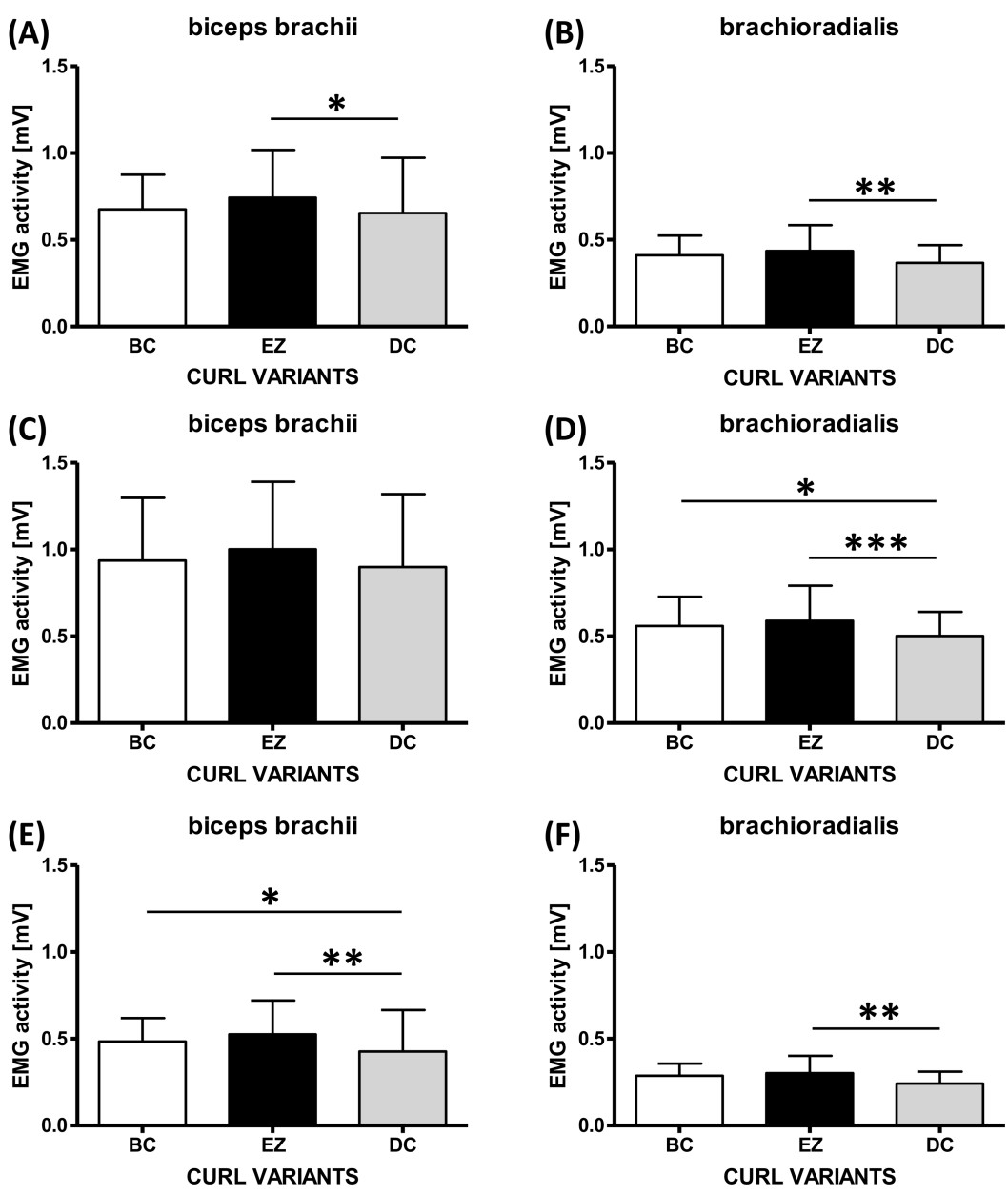

**Figure 4 EMG activity.** EMG activity of the biceps brachii and brachioradialis during (A–B) the whole range of motion, (C–D) the concentric phase, and (E–F) the eccentric phase (C). (*$p < 0.05$, **$p < 0.01$).

although there were no statistically significant differences. Conversely, the variation of curls on BR muscle activity during the concentric phase resulted to be statistically significant when comparing the EZ to the DC as well as the BC to the DC.

The eccentric phase showed lower EMG activity with respect to the concentric phase for both muscles investigated. Specifically, EZ and BC variants elicited a statistically significant higher EMG activity of the BB muscle with respect to the DC variant. BR muscle activity was found to be significantly higher only in the EZ variant compared to the DC.

As the comparison was made between two bilateral movements (BC and EZ) and one unilateral movement (DC), our results could have been affected by the phenomenon of the bilateral deficit (*Howard & Enoka, 1991*) which showed a higher EMG activity in the unilateral movement (DC). Nevertheless it seems that this was not the case in our study. Most it is possible that our findings were not affected by the bilateral facilitation (*Botton et al., 2016*) as all participants were equally familiar with both the BC or EZ and the DC variants.

Therefore, the anatomical aspects of these two muscles have to be taken into account when interpreting the results. Biceps brachii is a robust forearm supinator and an elbow flexor while brachioradialis, the most superficial muscle of the forearm considering the radial part, acts as an elbow flexor (*Williams et al., 1989*). Wire EMG analysis suggested that these two muscles, together with the brachialis, differ in their flexor activity depending on the three positions of the forearm: prone, semiprone and supine (*Basmajian & De Luca, 1978*). During the execution of the three exercise variants in the present study, the forearms assumed different positions through the dynamic elbow flexion movement. In the BC, both forearms are supined throughout the performance of the exercise while in the EZ barbell curl they assume an intermediate position very close to a semiproned position. In the DC, the forearm is semiproned at the initial phase of the repetition, after which it assumes a supine position at approximately 90° of elbow flexion until the end of the concentric phase. The almost semiproned forearm position during the EZ barbell curl could explain the higher muscle activity in this variant. Accordingly to *Basmajian & De Luca (1978)*, they reported that the biceps brachii, brachialis and brachioradialis act maximally when the weight is lifted by means of flexing the elbow throughout a semiproned forearm position.

Although significant differences were found between variants, some limitations of the present study need to be acknowledged. First, even if widely employed in kinesiologic and sport applications it has been demonstrated that bipolar surface EMG signals can be influenced by the thickness of the subcutaneous tissue layers, electrode size and shape, spatial filter transfer function, and interelectrode distance (*Farina, Cescon & Merletti, 2002*). Moreover the consensus on the electrode placement is still debated (*Mesin, Merletti & Rainoldi A. Surface, 2009*). Second, the present work did not assess EMG activation of the brachialis muscle. Even if it is still unclear whether its activity can be accurately assessed with surface electrodes at high levels of muscular contraction (*Staudenmann & Taube, 2015*), its exclusion in the analysis of the three variants of curl has to be acknowledged as a limitation to the present study since this muscle is one of the main contributor during elbow flexion. Third, a different ROM was reported in the three exercises variants. Nevertheless, the presence of little differences in term of ROM (approximately less than 6° ) was specific to the technical execution of each variant of curl and thus standardizing its value could have prevented the participants optimally performing the exercises.

## CONCLUSION

We can conclude that the EZ barbell curl was the most effective variant considering the overall EMG activity of the BB and BR. On the other hand, the DC variant was found to

be less effective, while the BC variant could be placed in an intermediate position but with an activation closer to the EZ than to the DC variant for the two muscles investigated.

Our findings suggested that the EZ barbell curl may be preferred to DC considering the whole phase of the repetition and the eccentric phase both for BB and BR. The small difference between the BC and EZ variants with regards to the EMG activity of the BB and BR, makes the choice between these two variants purely a matter of subjective comfort related to the handgrip position.

## ACKNOWLEDGEMENTS

The authors would like to thank all of the participants that took part in the study and Dr. Luqman Aziz for reviewing the English style.

### Funding

The authors received no funding for this work.

### Competing Interests

The authors declare there are no competing interests.

### Author Contributions

- Giuseppe Marcolin conceived and designed the experiments, performed the experiments, analyzed the data, prepared figures and/or tables, authored or reviewed drafts of the paper, approved the final draft.
- Fausto Antonio Panizzolo analyzed the data, authored or reviewed drafts of the paper, approved the final draft.
- Nicola Petrone contributed reagents/materials/analysis tools, approved the final draft.
- Tatiana Moro prepared figures and/or tables, authored or reviewed drafts of the paper, approved the final draft.
- Davide Grigoletto performed the experiments, contributed reagents/materials/analysis tools, approved the final draft.
- Davide Piccolo performed the experiments, analyzed the data, approved the final draft.
- Antonio Paoli conceived and designed the experiments, contributed reagents/materials/analysis tools, authored or reviewed drafts of the paper, approved the final draft.

### Human Ethics

The following information was supplied relating to ethical approvals (i.e., approving body and any reference numbers):

The study was approved by the ethical committee of the Department of Biomedical Sciences, University of Padova (number HEC-DSB12/16).

### Data Availability

The raw data are provided in a Supplemental Information 1.

## Supplemental Information

Supplemental information for this article can be found online at http://dx.doi.org/10.7717/peerj.5165#supplemental-information.

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

# PeerJ

different push-up variants. *Journal of Athletic Training* **50(11)**:1126–1132 DOI 10.4085/1062-6050-50.9.09.

**Mesin L, Merletti R, Rainoldi A. Surface EMG. 2009.** The issue of electrode location. *Journal of Electromyography and Kinesiology* **19(5)**:719–726 DOI 10.1016/j.jelekin.2008.07.006.

**Oliveira AS, Gonçalves M. 2009.** Positioning during resistance elbow flexor exercise affects electromyographic activity, heart rate, and perceived exertion. *Journal of Strength & Conditioning Research* **23(3)**:854–862 DOI 10.1519/JSC.0b013e3181a00c25.

**Oliveira LF, Matta TT, Alves DS, Garcia MAC, Vieira TMM. 2009.** Effect of the shoulder position on the biceps brachii EMG in different dumbbell curls. *Journal of Sports Science and Medicine* **8(1)**:24–29.

**Paoli A, Marcolin G, Petrone N. 2009.** The effect of stance width on the electromyographical activity of eight superficial thigh muscles during back squat with different bar loads. *Journal of Strength & Conditioning Research* **23(1)**:246–250 DOI 10.1519/JSC.0b013e3181876811.

**Paoli A, Marcolin G, Petrone N. 2010.** Influence of different ranges of motion on selective recruitment of shoulder muscles in the sitting military press: an electromyographic study. *Journal of Strength & Conditioning Research* **24(6)**:1578–1583 DOI 10.1519/JSC.0b013e3181d756ea.

**Signorile JF, Kwiatkowski K, Caruso JF. 1995.** RB. Effect of foot position on the electromyographycal activity of the superficial quadriceps muscles during the parallel squat and knee extension. *Journal of Strength & Conditioning Research* **9(3)**:182–187.

**Signorile JF, Zink AJ, Szwed SP. 2002.** A comparative electromyographical investigation of muscle utilization patterns using various hand positions during the lat pull-down. *Journal of Strength & Conditioning Research* **16(4)**:539–546.

**Staudenmann D, Taube W. 2015.** Brachialis muscle activity can be assessed with surface electromyography. *Journal of Electromyography and Kinesiology* **25(2)**:199–204 DOI 10.1016/j.jelekin.2014.11.003.

**Williams PL, Warwick R, Dyson M, Bannister LH. 1989.** *Gray's anatomy*. Edinburgh: Churchill livingstone.