# Peer review of "Differences in electromyographic activity of biceps brachii and brachioradialis while performing three variants of curl"

_PeerJ, doi:10.7717/peerj.5165_

## Round 0.1 · original submission · Major Revisions

I apologize for the delay, but it was very difficult to obtain reviewers for this manuscript. I have now received two reviews. One reviewer suggests rejection and the other major revisions. Having considered the feedback, it is clear to me that your work needs a profound revision before it can be considered for publication. Reviewer 1 suggests to obtain data from the triceps brachii and posits some comments about your EMG data. Reviewer 2 is concerned about your methodology and find several problematic points: the comparison of 2 bilateral movements to a unilateral; the differences in range of motion between the DC and BC variants; lack of information about which arm was used; a potential completely counterbalanced design; lack of information about the instructions with regard to movement intent in regards to velocity during the concentric and eccentric phases. I think that all of these are very important observations that need to be carefully addressed.

I will appreciate a consideration of all issues raised by both reviewers and a point-by-point response to the critical reviews you received.

Reviewer 1 ·

Basic reporting

Basic reporting is fine. I have no comments which could improve the quality of this section.

Experimental design

In my opinion the activity of the brachialis muscle, as the main and strongest flexor is very important. Furthermore, the activity of the antagonist muscles (triceps) should be recorded. This would add to the story.
Why are the EMG data not normalized?
The authors should show representative EMG data.
Surface EMG has a lot of limitations. This should be stated in the manuscript.

Validity of the findings

While the findings are robust, the significance and novelty is poor.

Additional comments

Minor comments: Replace "raw EMG" with "interference EMG".

·

Basic reporting

For comments on the English, please see the general comments section where this issue is addressed. In brief, there are many small grammatical errors throughout the manuscript and an English native speaker is needed to refine the manuscript. The literature references, background, and context provided seem to be adequate as is the professional article structure. The manuscript also appears to be self-contained with relevant results to the hypotheses.

Experimental design

The experimental design appears to be adequate if the main goal of the research is to provide practical information for bodybuilders or people training for general fitness. The research question and purpose are defined, fills a small gap in the literature, rather novel, and data seem to have been collected carefully. However, there are a few methodological details that are questionable and more detail needs to be provided on some of the methods.

Validity of the findings

Overall, I think the findings are valid if the main goal of the research is to provide practical information for bodybuilders or people training for general fitness. There are a few methodological issues that could be viewed as problematic (see general comments section). However, I do not think they are major enough to influence the main goal of providing information on muscle activation for practical training purposes. Finally, there are several issues regarding methodology and discussion of results that the authors need to address (see General Comments section).

Additional comments

The purpose of the present study was to determine if differences exist in the magnitude EMG activity between three variations of bicep curls (straight bar curl (BC), undulated bar curl (EZ), and alternate dumbbell curl (DC)) that are commonly used in bodybuilding and fitness training programs. Twelve healthy male participants (age: 25 ± 5 years, body mass: 77 ± 9 kg, height: 183 ± 6 cm) with at least three years resistance training experience (three lifting training sessions per week) participated in the study. The experimental design involved performance of a one repetition maximum (1-RM) for each of the three types of bicep curls by means of a submaximal estimation method. A week later, subjects performed 1 set of 10 repetitions with 65% of maximum with each type of bicep curl in random order with 4 minutes of rest between each exercise/set. During these sets, EMG activity was measured from the biceps brachii (BB) and the brachioradialis (BR) muscles of the right arm. EMG activity was quantified for the total rep, the concentric phase, and the eccentric phase for 8 of the 10 repetitions (first and last repetition not quantified as mechanically these are different). Finally, the range of motion was quantified with a goniometer.

The main findings were: 1) EMG activity was significantly higher for the BB and BR muscles for the EZ condition compared to the DC condition during the entire repetition; 2) For the concentric phase, BR activity was lower in the DC condition compared to the EZ and BC conditions; 3) For the eccentric phase, BB EMG activity was higher in the EZ and BC conditions compared to the DC conditions; and 4) Range of motion was greater in the BC condition compared to the DC condition.

Overall, the manuscript seems to have been written by authors with English as a second language. The authors should have an English speaker edit the manuscript as there are small grammatical errors in many sentences. While the authors did a commendable job on the writing portion of the paper considering it is their second language and far better than I could ever do with a second language, there are numerous sentences that need to be rewritten due to the grammar. However, the authors did do a good job of not having typographical and other types of errors as I only say perhaps a few of these. In regards to the grammatical errors there are too many to point out individually but a native English speaking writer should be able to easily fix these errors as most are very minor. This topic addressed seems to be somewhat novel as to my previous studies have not investigated differences in muscle activation due to the EZ curl bar, despite the widespread use of this bar specifically for bicep curls for many decades. The focus of the research seems to be appropriate for PeerJ and of interest to many readers of the journal. At the present time, there appear to be no fatal flaws in the paper, however, I have a number of concerns, clarifications, and changes the authors should address before the manuscript can be considered for publication.

Major Points:

1. The comparison of 2 bilateral movements to a unilateral one could be viewed as problematic. The neuromuscular mechanisms underlying the control and activation of bilateral and unilateral movements can be quite different. Most importantly, the phenomenon of the bilateral deficit or bilateral facilitation (Henry and Smith 1961, Krol 1965, Ohtsuki 1984, Howard & Enoka 1991, Botton et al. 2013) was not mentioned and could potentially confound the results. Due to the bilateral deficit, one would think that the unilateral movement should have had higher EMG but this was not the case. Perhaps this could have been due to a bilateral facilitation if the subjects had mainly performed bilateral bicep exercises throughout their prior training careers. The authors definitely need to add these points to the discussion and if possible ask the subjects about their prior training history and report the results. In summary, this could be a methodological issue but it can be addressed by including it in the discussion and surveying the subjects if possible. Nonetheless, for practical training purposes the observations of the differences between the exercises are still valuable for practical applications so I do not see this problem as a fatal flaw.

2. The differences in range of motion between the DC and BC variants could be viewed as problematic also. Personally, I don’t believe that this statistically significant difference was great enough to have a major physiological effect or influence the overall results considering how the data was analyzed. Nonetheless, the authors probably should add to the discussion more information on why this one difference in range of motion likely did not have a big influence on the results.

3. The authors state that the right arm was used, but they should also report on how many of the subjects were right hand dominant. I assume most or all were. Although I really don’t think the exact number of right and left handed individuals would influence the overall results, this sort of information should be reported in papers involving the upper limbs.

4. I think the authors should have used a completely counterbalanced design instead of randomization. This likely doesn’t matter if there wasn’t a major bias due to the randomization in the order of exercises performed. However, with 6 possible order combinations due to there being 3 conditions and 12 subjects recruited, the authors could have completely counterbalanced the design. The use of a 4 minute rest interval probably mostly mitigated any fatigue effects, but it still would have been better to counterbalance.

5. More detail needs to be provided in the Methods section regarding the instructions to the subject and to how the sets of repetitions of the various curls were performed. Most importantly, what were the instructions with regard to movement intent in regards to velocity during the concentric and eccentric phases? This information is not provided. It would have been best to perform the eccentric over a set time span (i.e. 2 seconds) and the concentric as fast as possible. Since the weights were normalized to 65% of maximum for each exercise in this single joint movement, I doubt there were major velocity differences between the three movements as the subjects likely did the movements at nearly the same velocity subconsciously and due to the normalized load. However, if the authors have this velocity information perhaps they should provide it along with what the instructions were to performance. Finally, more detail should be provided on how posture or possible cheating (moving at the waist etc) was controlled for or monitored.

---

## Round 0.2 · accepted · Accept

Thank you for taking into account the suggestions of our reviewers. I think that this manuscript is ready for publication.

·

Basic reporting

The authors have done a very good job of responding to all my previous concerns and have made the associated changes to the manuscript. One slight concern from before was the English, which has now been greatly improved. There are still a few spots where the English could be slightly improved. For instance, the part of the title that says "three variants of curl" would probably be improved if it read "three variants of bicep curls" or "three variations of the bicep curl". Overall, I think the article is ready for publication.

Experimental design

No comment.

Validity of the findings

No comment.

Additional comments

The authors have done a very good job of responding to all my previous concerns and have made the associated appropriate changes to the manuscript. One slight concern from before was the English and this has now been greatly improved. There are still a few spots where the English could be slightly improved. For instance, the part of the title that says "three variants of curl" would probably be improved if it read "three variants of bicep curls" or "three variations of the bicep curl". Overall, I think the article is ready for publication.